# The Hsf1-sHsp cascade has pan-antiviral activity in mosquito cells
Jieqiong Qu [1,2], Michelle Schinkel [1,2], Lisa Chiggiato [1], Samara Rosendo Machado [1], Gijs J. Overheul[1], Pascal Miesen [1] & Ronald P. van Rij [1] ✉

*Aedes* mosquitoes transmit pathogenic arthropod-borne (arbo) viruses, putting nearly half the world's population at risk. Blocking virus replication in mosquitoes is a promising approach to prevent arbovirus transmission, the development of which requires in-depth knowledge of virus-host interactions and mosquito immunity. By integrating multi-omics data, we find that heat shock factor 1 (Hsf1) regulates eight small heat shock protein (sHsp) genes within one topologically associated domain in the genome of the *Aedes aegypti* mosquito. This Hsf1-sHsp cascade acts as an early response against chikungunya virus infection and shows pan-antiviral activity against chikungunya, Sindbis, and dengue virus as well as the insect-specific Agua Salud alphavirus in *Ae. aegypti* cells and against chikungunya virus and O'nyong-nyong virus in *Aedes albopictus* and *Anopheles gambiae* cells, respectively. Our comprehensive in vitro data suggest that Hsf1 could serve as a promising target for the development of novel intervention strategies to limit arbovirus transmission by mosquitoes.

A*edes* mosquitoes are the primary vectors of arthropod-borne (arbo) viruses[1] that pose significant health and economic burden to humans[2–4]. Dengue virus (DENV) alone causes approximately 400 million infections and 20,000 deaths annually[5]. Outbreaks of chikungunya virus (CHIKV) in Europe and the Americas have also garnered significant attention[6–11]. With the continued resurgence of arboviruses and their unprecedented geographical spread over the past years, there is a pressing need for novel strategies to prevent and mitigate future outbreaks.

Arbovirus transmission occurs primarily through a horizontal cycle between blood-feeding arthropod vectors, such as mosquitoes, and vertebrate hosts. Female mosquitoes ingest viruses during a blood meal from an infected host and then transfer viruses into a naïve host through biting[12,13]. Blocking virus transmission at the mosquito stage, rather than solely targeting the virus in human hosts, constitutes a promising, alternative approach to prevent arbovirus spread[14]. The development of such approaches requires in-depth knowledge of mosquito antiviral immunity.

Our current understanding of the insect immune system is largely informed by studies in *Drosophila melanogaster*[15–18], where the RNA interference (RNAi) pathway serves as a major antiviral response that degrades viral RNA and inhibits replication[19–21]. Innate immune signaling pathways, specifically the NF-κB (Toll and IMD) and JAK-STAT pathways, have also been implicated in antiviral defense[16,22]. However, *Drosophila* does not recapitulate mosquito specific processes such as virus acquisition and transmission via blood-feeding[12,13,15,22]. It is therefore crucial to study immunity against arboviruses in relevant species, such as *Ae. aegypti*, the primary vector of the urban virus transmission cycle[23,24].

RNAi has been identified as a major antiviral response in *Ae. aegypti* mosquitoes, for example, against CHIKV[25], Zika virus (ZIKV)[26,27], DENV[28,29], and Sindbis virus (SINV)[30]. In contrast, the role of classical innate immune pathways is less well-defined and seems to be virus dependent[31–36]. For instance, ZIKV infection activates the Toll pathway, while DENV infection activates both the Toll and JAK-STAT pathways[31–33]. Activation of the Toll and JAK-STAT pathways by depleting pathway-specific negative regulators Cactus and PIAS, respectively, suppresses ZIKV infection[31]. On the other hand, activation of the IMD pathway by silencing of Caspar did not affect ZIKV infection[31], but impaired SINV replication[34]. Additionally, stimulation with either heat inactivated *Escherichia coli* to activate the JAK-STAT and IMD pathways or *Bacillus subtilis* to activate Toll pathway had no effect on CHIKV replication[25], suggesting unknown antiviral mechanisms yet to be discovered.

To investigate the transcriptional responses of *Ae. aegypti* to viral infection, we conducted a time-course RNA-seq experiment upon CHIKV infection. Our combined analyzes of RNA-seq and chromatin profiling data revealed an early-responsive cascade, consisting of heat shock factor 1 (Hsf1) driving the expression of eight small heat shock proteins (sHsps). This Hsf1-sHsp cascade displayed conserved pan-antiviral activities in in vitro experiments on three vector mosquitoes. As such, our data suggest that

[1]Department of Medical Microbiology, Radboud University Medical Center, Nijmegen, The Netherlands. [2]These authors contributed equally: Jieqiong Qu, Michelle Schinkel. ✉e-mail: Ronald.vanRij@radboudumc.nl

 1

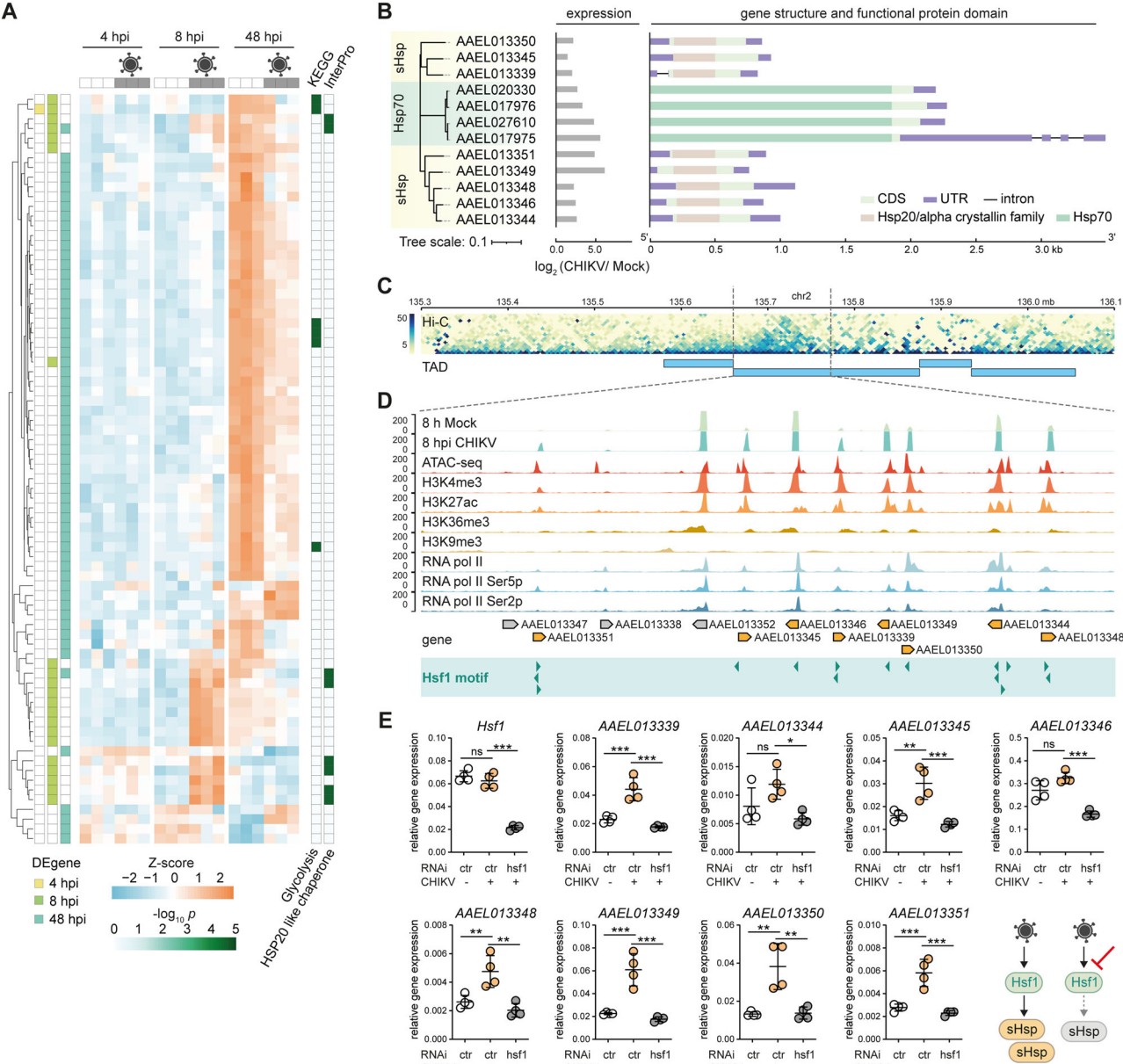

**Fig. 1 | Hsf1 activates sHsp expression during the early stage of CHIKV infection.**
**A** Hierarchical clustering of differentially expressed (DE) genes at 4, 8, and 48 hpi of mock or CHIKV infection (MOI = 5) in *Aedes aegypti* Aag2 cells, as quantified from RNA-seq data ($n = 3$ biological replicates). DEgenes ($p < 0.05$) at each time point are color labeled on the left; KEGG pathway and InterPro enrichment is shown on the right. Z scores were calculated based on log-transformed Fragments per Kilobase per Million mapped reads (FPKM). **B** Gene structure and protein domain analysis of heat shock proteins upregulated at 8 hpi of CHIKV infection. A neighbor-joining tree was constructed based on amino acid sequences (left). The $\log_2$ fold changes in gene expression were quantified by RNA-seq (middle). The gene structure and InterPro protein domains are shown along the gene body (right). **C** Hi-C interaction track with

topologically associated domains (TADs) shown as horizontal blue bars ($p < 0.05$). **D** Genome browser tracks showing RNA-seq data of mock and CHIKV infected Aag2 cells and chromatin profiling data from uninfected Aag2 cells (ATAC-seq, histone marks and RNA pol II ChIP-seq) within the indicated TAD. Eight upregulated *sHsp* genes at 8 hpi are highlighted in orange, other genes in gray. Predicted Hsf1 motifs are shown as green arrowheads at the bottom. **E** Expression of the indicated genes in mock (-) or CHIKV (MOI = 5) infected Aag2 cells upon RNAi-mediated silencing of *Hsf1* at 8 hpi. Luciferase dsRNA served as a non-targeting control (ctr). RT-qPCR was performed in biological quintuplicates and shown as dots, with horizontal lines and the bars indicating mean ± SD. Symbols denote statistical significance in unpaired *t*-tests (ns, not significant, $*p < 0.05$; $**p < 0.01$; $***p < 0.001$).

---

Hsf1 serves as a promising target for the development of new intervention strategies to control arbovirus transmission by mosquitoes.

## Results
### The Hsf1-sHsp cascade acts as an early response to CHIKV infection

To uncover novel components involved in antiviral defense in *Aedes* mosquitoes, we performed a time-course RNA-seq experiment to determine differentially expressed genes (DEgenes) upon CHIKV infection of

*Ae. aegypti* Aag2 cells at 4, 8, and 48 h post-infection (hpi) (Fig. 1A and Supplementary Data 1). The CHIKV strain that emerged during the 2005–2006 outbreaks, which acquired a mutation (A226V) in the E1 envelope glycoprotein, was used in this study[37]. A multiplicity of infection (MOI) of 5 was used to ensure that the vast majority of cells are synchronously infected. Pilot RT-qPCR indicated that CHIKV RNA levels strongly increased from 4 hpi onwards (Supplementary Fig. 1A), mirroring the dynamics of CHIKV RNA reads in our RNA-seq data (Supplementary Fig. 1B). At 4 hpi, only a single DEgene was detected

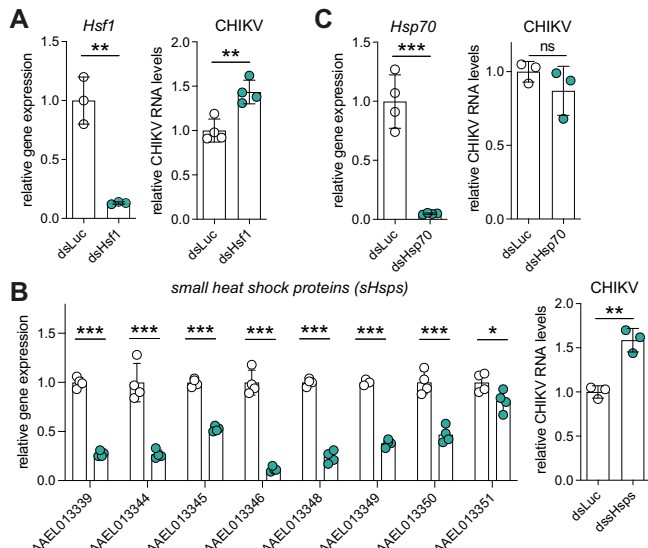

**Fig. 2 | The Hsf1-sHsp axis has antiviral activity against CHIKV. A–C** RNAi silencing of *Hsf1* **A**, *sHsps* **B**, *Hsp70* (*AAEL017975*) **C**, and quantification of CHIKV RNA copies. dsRNA targeting the highly conserved Hsp20 domain was used to knock down all eight sHsp genes simultaneously in **B**. RT-qPCR was used to assess target gene knockdown in uninfected Aag2 cells (left) and to quantify CHIKV RNA copies at 8 hpi (MOI = 5) after target gene knockdown (right) in biological triplicate or quadruplicate, with luciferase dsRNA (dsLuc) serving as a non-targeting control. Individual data are shown as dots, with bars indicating mean ± SD. Symbols denote statistical significance in unpaired *t*-tests (ns, not significant, *$p < 0.05$; **$p < 0.01$; ***$p < 0.001$).

(*AAEL000080*, encoding a putative phosphoenolpyruvate carboxykinase), which was downregulated at 8 hpi (Fig. 1A). We observed major transcriptional changes at 8 hpi and 48 hpi, which we termed the early and late responses to CHIKV infection, respectively. The late response was more pronounced, with eight upregulated and 51 downregulated genes (Fig. 1A), and KEGG pathway analysis suggested that downregulated DEgenes were mainly involved in glycolysis, which we have reported before[38]. In contrast, the early response was modest, with two downregulated and 19 upregulated genes, which were enriched for genes encoding HSP20-like chaperones (Fig. 1A). Protein domain analysis revealed four genes encoding heat shock protein 70 (Hsp70) and eight genes encoding small heat shock proteins (sHsps) (Fig. 1B), which function to prevent protein aggregation and help cells to survive stressful conditions[39].

While Hsp70 is involved in distinct steps of the viral cycle of dengue and Zika virus in mosquito cells, including entry, RNA replication, and virion assembly[40,41], the role of sHsps during virus infection is largely unknown[42,43]. Inspecting the genomic distribution of these sHsp genes, we found that all eight sHsps are encoded within one Topologically Associated Domain (TAD) deduced from Hi-C seq data[44] (Fig. 1C), suggesting that these sHsp genes are likely to be coordinately regulated by the same cis-regulatory network[45,46]. To further study the regulation of sHsp gene expression within this TAD, we mapped our chromatin profiling data from Aag2 cells[47], and performed motif scan within open chromatin regions defined by the Assay for Transposase-Accessible Chromatin with sequencing (ATAC-seq). We found the DNA binding motif of the transcription factor heat shock factor 1 (Hsf1) to be highly enriched at promoter regions of these eight sHsp genes, but not at other genes within the same TAD (Fig. 1D). These regions were also marked by active histone marks H3K4me3 and H3K27ac, as well as RNA polymerase II occupancy, indicating active transcription[48,49]. These findings together suggest that Hsf1 is the transcription factor that regulates sHsp gene expression. To experimentally test this hypothesis, we RNAi silenced *Hsf1*, and indeed, the induction of all eight *sHsps* upon CHIKV infection was significantly diminished (Fig. 1E). Hence, the Hsf1-sHsp cascade functions as an early transcriptional response to CHIKV infection in Aag2 cells.

## The Hsf1-sHsp cascade displays anti-CHIKV activity

To explore the antiviral potential of the identified Hsf1-sHsp axis, we silenced *Hsf1* using RNAi and observed elevated CHIKV RNA levels at 8 hpi (Fig. 2A). sHsp genes share high similarity at the nucleotide level, we therefore used two sets of dsRNA targeting the conserved Hsp20 domain to knock down all eight *sHsp* genes and observed significantly increased CHIKV RNA copies at 8 hpi (Fig. 2B). In contrast, knockdown of *Hsp70* did not affect CHIKV replication (Fig. 2C), although we cannot exclude that Hsp70 may affect other stages of the viral life cycle[40,41]. These results collectively suggest that the Hsf1-sHsp axis has antiviral activity against CHIKV infection.

We then modulated the activity of Hsf1 with either the small molecule inhibitor KRIBB11[50] or the activator hsfa1[51]. KRIBB11 inhibits Hsf1-dependent recruitment of positive Transcription Elongation Factor b to the promoters of target genes, thereby blocking gene activation[50]. KRIBB11 treatment significantly downregulated most *sHsp* genes after 12 h, although this effect got weaker after 24 h (Fig. 3A-B). The precise mechanism by which hsfa1 activates Hsf1 remains unclear[51]. We observed strong activation of *sHsp* genes following hsfa1 treatment at both 12 h and 24 h. To further validate our findings, we included another compound, Direct Targeted Hsf1 InhiBitor (DTHIB), reported as a direct human Hsf1 inhibitor by stimulating the degradation of nuclear Hsf1[52]. Unexpectedly, DTHIB treatment activated *sHsp* expression in Aag2 cells at both 12 h and 24 h, similar to hsfa1 (Fig. 3A-B). Thus, DTHIB acts as an Hsf1 activator rather than an inhibitor in mosquito cells.

Subsequently, we assessed the impact of Hsf1 modulation on CHIKV infection using the aforementioned compounds (Fig. 3C,D). At 12 hpi, KRIBB11 treatment did not affect *Hsf1* expression, but led to more than 5-fold increase of viral RNA levels in Aag2 cells (Fig. 3C), a trend that remained concordant at 24 hpi (Fig. 3D). While the expression of most *sHsp* genes was significantly reduced with KRIBB11 treatment, counterintuitively, we observed de-repression or even activation of some *sHsps*, such as *AAEL013346*, *AAEL013349*, and *AAEL013351* at 12 hpi, which is likely a secondary effect of increased CHIKV replication (Fig. 3C). Hsfa1 treatment significantly increased *Hsf1* expression and strongly activated all eight *sHsp* genes, leading to about a 25-fold reduction in CHIKV copies at 12 hpi and a 10-fold reduction at 24 hpi (Fig. 3C, D). In contrast, DTHIB did not affect *Hsf1* expression at 12 hpi, but still elevated *sHsp* gene expression and consequently repressed CHIKV replication (Fig. 3C). These findings align with our RNAi experiments, reinforcing the antiviral role of sHsps.

## The Hsf1-sHsp cascade affects an early, post-entry stage of the viral replication cycle

To determine which stage of the CHIKV life cycle is affected by the Hsf1-sHsp cascade, we performed time-of-addition assays in Aag2 cells (Fig. 4A). DMSO treated cells were harvested at each indicated time point to establish a reference virus replication curve. When the Hsf activator hsf1a was added at time points before inoculation and up to 4 hpi, CHIKV replication was strongly inhibited (Fig. 4B). This observation suggests that hsfa1 affects a critical step occurring around 4 hpi but does not impact earlier stages, such as the virus attachment or entry step, which typically occur within 1 hour after inoculation[53]. After 4 hpi, both positive and negative sense viral RNA copies became detectable, and the inhibitory effect of hsf1a on CHIKV replication began to gradually diminish (Fig. 4C). This indicates that the Hsf1-sHsp cascade interacts with the early stage of CHIKV RNA replication but does not affect subsequent stages. To validate this time window, we performed a time-of-addition assay with the inhibitor KRIBB11 at 2, 4, 6, 8, 10 hpi, which showed the same pattern: a gradual loss of activation of CHIKV replication after 4 hpi (Fig. 4D,E). Together, these results demonstrate that the Hsf1-sHsp cascade likely reduces early CHIKV replication.

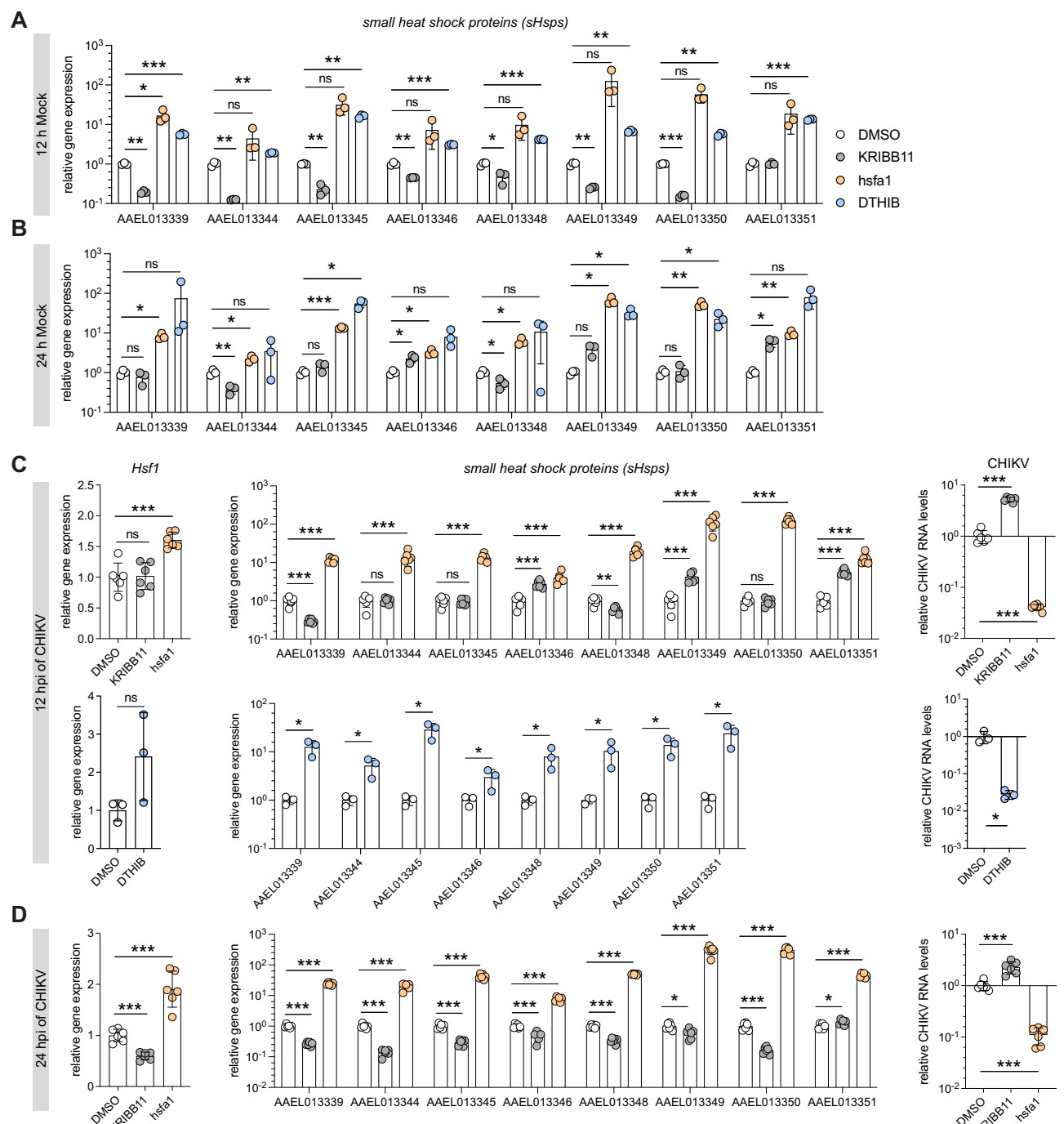

**Fig. 3 | Pathway modulation using small-molecule compounds confirms the antiviral activity of the Hsf1-sHsps axis. A**, **B** Effects of Hsf1 modulation using KRIBB11 (10 μM), hsfa1 (100 μM), and DTHIB (10 μM) on *sHsp* expression measured by RT-qPCR at 12 h **A** and 24 h **B**. DMSO was used as control. **C**, **D** Expression analysis of *Hsf1*, *sHsp* genes, and CHIKV RNA copies following treatment with KRIBB11 (10 μM), hsfa1 (100 μM), and DTHIB (10 μM) at 12 hpi **C** and 24 hpi **D** with CHIKV (MOI = 5), using DMSO as a control. RT-qPCR data are presented in biological triplicate, quadruplicate, or sextuplicate, displayed as dots, with bars indicating mean ± SD. Statistical significance in unpaired *t*-tests is denoted by symbols (ns, not significant; *$p < 0.05$; **$p < 0.01$; ***$p < 0.001$).

### The Hsf1-sHsp cascade showed pan-antiviral activity

We next investigated the effect of the Hsf1-sHsp cascade on replication of other alphaviruses in Aag2 cells, including SINV (AR339 strain), the insect-specific Agua Salud alphavirus (ASALV), and the flavivirus DENV (serotype 2). Consistently, KRIBB11 treatment led to increased virus replication for SINV and ASALV, whereas hsfa1 treatment significantly reduced virus replication for SINV, ASALV and DENV (Fig. 5A–C). We extended our findings to in vitro experiments in other mosquito species and confirmed the antiviral property of the Hsf1-sHsp cascade in several natural virus-vector combinations[54,55], such as CHIKV in *Aedes albopictus* U4.4 cells (Fig. 5D) and C6/36 cells (Fig. 5E), and O'nyong-nyong (ONNV) in *Anopheles gambiae* Mos55 cells (Fig. 5F). The fact that the heat shock response was antiviral in both U4.4 cells and the RNAi-deficient C6/36 cells indicates that the Hsf1-sHsps cascade functions independently of RNAi. While the in vivo relevance remains to be determined (Supplementary Fig. 2), our results collectively suggest that the Hsf1-sHsp pathway displays a pan-antiviral role in cell lines from all three tested mosquito species.

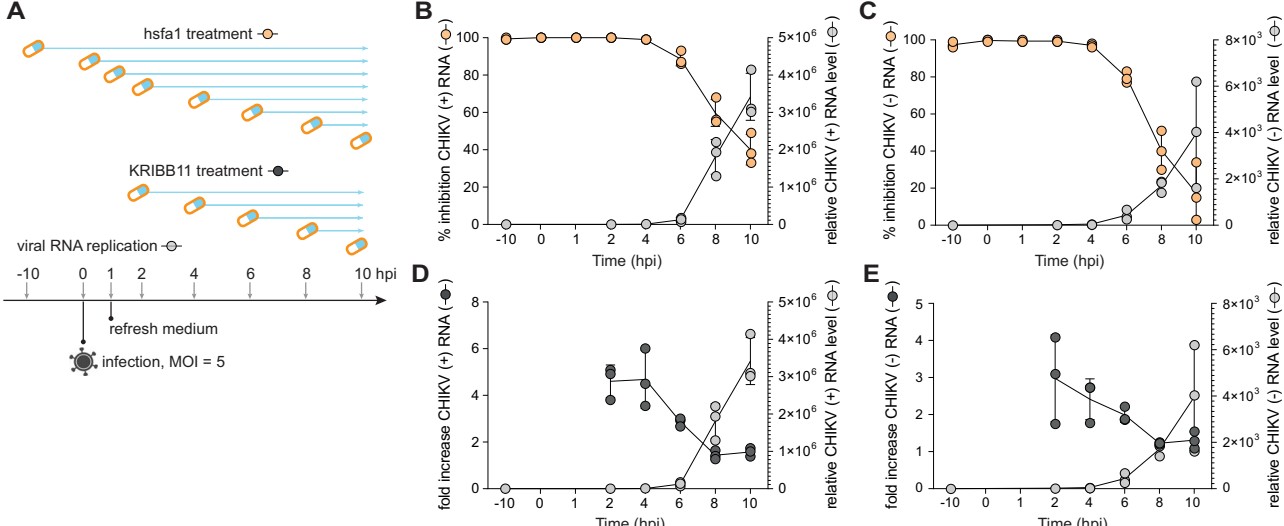

**Fig. 4 | The Hsf1-sHsp cascade affects the early stage of the CHIKV replication cycle. A** Schematic illustration of the time-of-addition assay during CHIKV infection (MOI = 5) in Aag2 cells. Cell culture medium was refreshed to wash away unbound virus particles at 1 hpi. DMSO (light gray, Mock) was used to establish a virus replication curve, with samples harvested at the indicated time points. hsfa1 (activator, orange) and KRIBB11 (inhibitor, dark gray) were added at specified time points, and cells were harvested at 10 hpi to assess their impact on CHIKV RNA levels. **B–E** The effect of hsfa1 (100 μM) **B–C** and KRIBB11 (10 μM) **D–E** on CHIKV replication was assessed using RT-qPCR on positive sense (+) viral RNA **B,D** and negative sense (−) viral RNA **C,E**. For comparative purposes, the replication curves of (+) and (−) viral RNA (light gray) have been duplicated in **B,D** and **C,E**, respectively. Data are presented as mean ± SD and individual data of three replicates. DMSO was used as the negative control to calculate the percent inhibition upon hsfa1 treatment **B–C** and the fold increase in CHIKV RNA levels upon KRIBB11 treatment **D–E**.

## Discussion

Our combined analysis of RNA-seq and epigenetic profiling data revealed an early-responsive Hsf1-sHsp cascade activated upon CHIKV infection, which we found to have broad in vitro antiviral activity against alphaviruses and a flavivirus in cell lines of three vector mosquito species: *Ae. aegypti*, *Ae. albopictus*, and *An. gambiae*. Overall, our work uncovers an antiviral cascade Hsf1-sHsp and identifies Hsf1 as a potential target for developing transmission-blocking strategies against arboviruses at the mosquito stage.

While RNAi is a well-established antiviral mechanism in mosquito vectors, it is becoming evident that inducible signaling pathways are of vital importance as well. In addition to the classical Toll, IMD, and JAK-STAT pathways, Wnt, mTOR, Ras/ERK, JNK pathways, and the Hsf1-sHsp cascade identified in this study, have been implicated in antiviral host defense[31,33,34,36,56–58]. Our time-course RNA-seq data revealed that the temporal activation of signaling pathways and associated genes is precisely regulated, with *sHsp* gene activation mainly restricted to the early stage of CHIKV infection and not detectable during later stages. In line with our findings, published RNA-seq data from an in vivo study of *Ae. aegypti* mosquitoes showed a similar pattern, with a higher induction of *sHsp* expression at the early stage (3 dpi) than at the late stage (7 dpi) of CHIKV infection[42]. In comparison to the well-studied role of Hsp70 at various stages of the viral cycle[40,41,59,60], few studies analyzed the role of sHsps in viral infection. This gap in research may be due to the transient activation patterns of sHsp genes during the early stages of virus infection. Their activation is highly dependent on the virus infection dynamics, making it challenging to capture this fleeting time window. In addition, arboviruses need to overcome tissue barriers to establish a persistent infection of the mosquito. Consequently, the infection state of different tissues or body parts can vary significantly. This may explain why only two studies reported the activation of sHsps[42,43], among the many transcriptome studies that have been conducted in the context of CHIKV infection in *Aedes* mosquitoes[36,42,43,61–64]. Single-cell RNA-seq has emerged as a powerful tool that may help to address the heterogeneity of sHsp activation in a tissue-specific or even cell-specific context.

To gain deeper insights into the regulation of *sHsp* genes, we performed integrative analyzes of 3D chromatin conformation (Hi-C seq) and chromatin profiling data (ATAC-seq and ChIP-seq) and identified Hsf1 as the upstream transcription factor that regulates expression of eight *sHsp* genes within one TAD. Targeting the transcription factor Hsf1, rather than individual downstream genes, is expected to be more effective for antiviral control. Together, our multi-omics analysis strategy demonstrates its power in discovering novel antiviral signaling pathways, which can be applied to further investigate virus-vector interactions in different tissues, in other mosquito species, and in other arbovirus infections.

The mechanism underlying the antiviral activity of the Hsf1-sHsp cascade remains largely elusive. Our time-of-addition assays suggest that the response inhibits an early post-entry stage of the viral replication cycle. As Hsf1 activation led to a broad antiviral effect in three vector mosquitoes, it is likely that the Hsf1-sHsp cascade interacts, either directly or indirectly, with a common viral protein feature or a conserved host protein complex. It has been suggested that sHsps maintain protein homeostasis by binding proteins in non-native conformations, thereby preventing substrate aggregation[65]. The fact that a wide range of proteins is protected from precipitation suggests that sHsps may act rather promiscuously, similarly to the molecular chaperone Hsp70[65–69]. It will be of great interest to profile sHsp associated proteins using proteomic approaches to identify sHsp client proteins and shed light on its antiviral mechanism.

To sum up, we identified the Hsf1-sHsp cascade as an antiviral mechanism in cell models of different vector mosquitoes. While the importance of the pathway for antiviral defense remains to be established in vivo, we propose Hsf1 as a potential target for developing arbovirus-blocking interventions. Considering the pan-antiviral activity of the Hsf1-sHsp cascade, our findings may open the way for the development of a broad-spectrum drug to prevent arbovirus transmission.

## Materials and Methods
### Cell lines

*Ae. aegypti* Aag2 cells were kindly provided by Raul Andino (University of California San Francisco), *Ae. albopictus* U4.4 cells by Gorben Pijlman (Wageningen University), and C6/36 cells by Sandra Junglen (Charité – Universitätsmedizin, Berlin). These cell lines were grown in Leibovitz's L-15 medium (Gibco 21083-027) supplemented with 10% fetal bovine serum

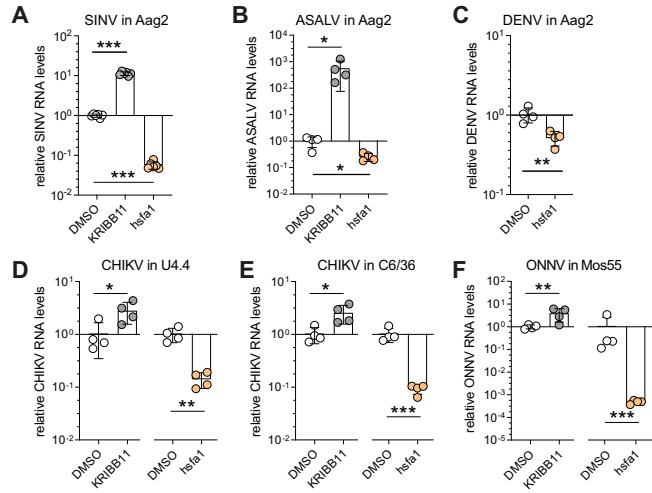

**Fig. 5 | Pan-antiviral activity of the Hsf1-sHsp axis. A–C** Viral RNA levels in *Aedes aegypti* Aag2 cells treated with KRIBB11 (10 μM) or hsfa1 (100 μM) at 12 hpi of Sindbis virus (SINV, MOI = 5) **A**, at 12 hpi of Agua Salud alphavirus (ASALV, MOI = 5) **B**, and at 72 hpi of Dengue virus (DENV, MOI = 0.01) **C**. **D–F** Viral RNA levels upon treatment with KRIBB11 (5 μM) and hsfa1 (100 μM) at 12 hpi of CHIKV (MOI = 5) in *Aedes albopictus* U4.4 cells **D**, at 12 hpi of CHIKV (MOI = 5) in *Dicer-2* deficient *Aedes albopictus* C6/36 cells **E**, at 48 hpi of o'nyong'nyong virus (ONNV, MOI = 0.5) in *Anopheles gambiae* Mos55 cells **F**. RT-qPCR was performed in biological quadruplicate or sextuplicate, shown as dots, with the bars indicating mean ± SD. Symbols denote statistical significance in unpaired *t*-tests (ns, not significant, \**p* < 0.05; \*\**p* < 0.01; \*\*\**p* < 0.001).

(Sigma F7524-500ML), 50 U/ml Penicillin, 50 μg/mL Streptomycin (Gibco 15070-063), 1x non-essential Amino Acids (Gibco 11140-035), and 2% Tryptose phosphate broth solution (Sigma T8159-100ML). *An. gambiae* Mos55 cells were kindly provided by Jason L. Rasgon (The Pennsylvania State University) and were grown in Schneider′s insect medium (Gibco 21720024) supplemented with 10% fetal bovine serum, 50 U/ml Penicillin, and 50 μg/mL Streptomycin. Mosquito cell lines were cultured at 28°C without $CO_2$. BHK-15 (kind gift of Jolanda Smit, University of Groningen) and BHK-21 cells (ATCC) were grown in Dulbecco's modified Eagle medium (Gibco 41965039), supplemented with 10% fetal bovine serum, 50 U/mL penicillin, and 50 μg/mL streptomycin. Mammalian cells were cultured at 37°C, with 5% $CO_2$.

### Viruses

The CHIKV expression plasmid encoding the Leiden synthetic wildtype strain (LS3, GenBank: JX911334.1) was kindly provided by Dr. M.J. van Hemert (Leiden University Medical Center) and viral RNA was obtained by in vitro transcription on linearized plasmids using T7 mMessage mMachine (Invitrogen AM1344)[37]. RNA was then transfected into BHK-21 to grow the infectious virus. SINV AR339 strain was obtained by transfecting in vitro transcribed RNA from linearized plasmid pTE3′ 2 J in BHK-21 cells[70]. ASALV (GenBank: MK959114), kindly provided by Sandra Junglen (Charité, Berlin), was originally isolated from the homogenate of mosquito pool MP416 and grown in C6/36 cells[71]. ONNV plasmid was kindly provided by Prof. Dr. Andres Merits (University of Tartu), in vitro transcribed, and grown in C6/36 cells[72]. DENV serotype 2 (DENV2, strain 16681) was provided by Beate Kümmerer (University of Bonn) and stocks were prepared on C6/36 cells. For the quantification of viral stocks, CHIKV, SINV, and ONNV were titrated by end-point dilution on BHK-21 cells, DENV2 was titrated on BHK-15 cells, and ASALV was titrated on C6/36 cells. A day before the titration, $1 \times 10^4$ mammalian or $5 \times 10^4$ C6/36 cells were seeded per well in a 96-well flat bottom plate. A 10-fold serial dilution of virus samples was added to the cells in quadruplicate for the titration. After an incubation of 5 days, cells were inspected for cytopathic effect. The virus titer was calculated according to the Reed and Muench method[73].

### Mosquitoes

*Ae. aegypti* mosquitoes (Black Eye Liverpool strain, obtained from BEI Resources, NR-48921), were reared at 28 °C, with 70% humidity and automated room lighting set at a 12:12 hours light/dark cycle. Larvae were fed with Tetramin Baby fish food (Tetra). Adult mosquitoes were fed with a 10% sucrose solution. Three-to-five-day-old female mosquitoes were used for experiments. For blood feeding experiments, mosquitoes were starved for 24 h prior to the experiment. CHIKV virus stock ($5 \times 10^8$ $TCID_{50}$/mL) and full human blood (Sanquin) was mixed at a 1:2 ratio to prepare the infectious blood meal with a final virus titer of $1.7 \times 10^8$ $TCID_{50}$/ml. DMEM medium was mixed in the same ratio with human blood to prepare the mock inoculum. Blood feeding was performed with the Hemotek system for half an hour. Afterwards, mosquitoes were anesthetized on a $CO_2$ pad. Only blood fed mosquitoes were selected and kept in culture until the indicated time points. Mosquitoes were tissue dissected as previously described[74]. Dissected body parts or tissues were homogenized in either 200 μl DMEM medium using a Precellys 24 homogenizer program 5 (Bertin technologies).

### RNAi in mosquito cells

PCR products flanked by T7 promoter sequences at both ends were in vitro transcribed by T7 polymerase to produce dsRNA using homemade T7 RNA polymerase. For the formation of double-stranded RNA, the reactions were heated to 90 °C for 10 min and then allowed to gradually cool to room temperature, and RNA was purified using the GenElute Mammalian Total RNA kit (Sigma RTN350) according to the manufacturer's protocol. Two sets of dsRNA targeting the Hsp20 domain were used to knock down *sHsp* expression (Fig. 2B) and two sets of dsRNA targeting the Hsp70 domain were used to knock down *Hsp70* (Fig. 2C). For detailed oligonucleotide information, see Supplementary Table 1.

Aag2 cells were seeded at a density of $1.5 \times 10^5$ cells/well in a 24-well plate, 24 h before the dsRNA transfection. For each target gene, a transfection mix containing 300 μl non-supplemented L-15 medium, 500 ng dsRNA, and 1.3 μl X-treme GENE HP DNA transfection reagent (Roche 6366546001) was prepared according to the manufacturer's recommendations. Per well, 100 μl of the transfection mix was added in a dropwise manner. Cell culture medium was refreshed at 4 h post-transfection. Three days post-transfection, cells were either harvested for evaluation of target gene knockdown or infection with CHIKV at an MOI of 5.

### RNA isolation and RT-qPCR

Aag2 cells were homogenized in RNA-Solv reagent (Omega Bio-Tek) and RNA extraction was performed according to the manufacturer's instructions. Briefly, 200 μl of chloroform was added to 1 mL RNA-Solv reagent and thoroughly mixed. After centrifugation, the aqueous phase was collected, and RNA was precipitated using isopropanol. This mix was incubated for 1 h at 4 °C, followed by centrifugation to pellet the RNA. Pellets were washed twice in 80% ethanol and dissolved in nuclease free water. RNA integrity was evaluated on ethidium bromide-stained agarose gels, and RNA concentration was determined using the Nanodrop ND-1000.

For RT-qPCR analyzes, DNaseI (Ambion)-treated RNA was reverse transcribed with random hexamers using the TaqMan Reverse Transcription Reagents (Invitrogen N8080234) and PCR amplified using the GoTaq qPCR system (Promega A6002) on a LightCycler 480 (Roche). In RNAi experiments, expression levels of target genes were normalized to the expression of the housekeeping gene *lysosomal aspartic protease* (*LAP*, *AAEL006169*), and fold changes were calculated using the 2-ΔΔCT method[75]. All primers are listed in Supplementary Table 1.

### RNA-seq library preparation and analysis

For RNA sequencing in Aag2 cells, control dsRNA (dsLuc) treated cells were infected with CHIKV at an MOI of 5. The dsRNA treatment was included in the procedure to allow a better comparison of the 4 hpi and 8 hpi samples to the 48 hpi datasets that was generated in parallel for a previously published study (PRJNA885496)[38].

TruSeq Stranded mRNA kit (Illumina 20020594) was used for library preparation from total RNA according to the manufacturer's protocol. The prepared libraries were quantified and controlled for sample quality using High Sensitivity DNA Chip on a DNA1000 Bioanalyzer (Agilent 5067-4626), and the libraries were analyzed on an Illumina HiSeq 4000 sequencer (GenomEast Platform).

After initial quality control by FastQC[76], raw sequence reads were aligned to the reference genome, *Ae. aegypti* LVP (VectorBase, release 56) and the plasmid CHIKV genome (LS3, GenBank: JX911334.1), using STAR[77] with default settings. Quantification of gene expression (FPKM) was performed with Cuffdiff[78]. R package DESeq2 was used for differential gene expression analysis (with adjusted $p$ value < 0.05)[79]. Genes were considered expressed if the mean of the DESeq2-normalized counts (basement) was higher than 20. The R package pheatmap was used to generate the heatmap for DEgenes upon CHIKV infection, which was based on z-scores of normalized gene expressions ($\log_{10}$ FPKM)[80]. DEgenes are listed in Supplementary Data S1.

### Bioinformatic analyzes

STRING was used for KEGG pathway and InterPro protein domain enrichment analysis[81]. Detailed protein domain analysis was performed using InterPro[82].

For 3D chromatin conformation analysis, TADs were defined by analyzing Hi-C data (GSE95797)[44] with HiCExplorer[83]. A cutoff of $p < 0.05$ was used, after correcting for multiple tests. pyGenomeTracks from the deepTools package[84] was used to visualize gene expression pattern, TADs, and previously published chromatin profiling data from Aag2 cells, including ATAC-seq, histone mark ChIP-seq and RNA pol II ChIP-seq (GSE210270)[47]. All genome browser tracks were normalized to FPKM (y-axis). Open chromatin regions were defined as ATAC-seq peaks called by MACS2 with standard settings and a $q$ value of 0.05[85]. Homer was used for DNA motif scan on ATAC-seq peaks[86].

### Compound experiment

KRIBB11, Hsf1a, and DTHIB were purchased from Axon Medchem (catalog numbers 2538, 1890, and 3412, respectively). All compounds were dissolved in DMSO to a final concentration of 10 mM (KRIBB11) or 100 mM (Hsf1a and DTHIB) and stored at 4 °C for use within one month. Mosquito cells were seeded at a density of $1.5 \times 10^5$ cells/ well in a 24-well plate. Cells at 50% confluency were treated with the indicated concentrations of compound or an equivalent volume of DMSO and immediately afterwards inoculated with virus at the indicated MOI. Cell lysates were collected at the indicated time points for RNA isolation and RT-qPCR. For the time-of-addition assay, the same procedures were followed, except that the medium was replaced at 1 hpi. For mosquito experiments, DTHIB was directly added to the (infectious) blood meal at a concentration of 10 μM and mixed well before blood feeding, with DMSO serving as a control.

### Statistics and reproducibility

RT-qPCR experiments in Figs. 1–5 were performed in biological triplicate, quintuplicate, sextuplicate, or octuplicate, with individual data shown as dot and the bars indicating mean ± SD. Statistical differences were examined by unpaired two-tailed $t$-tests. Nonparametric Mann–Whitney U and chi-squared tests were used for comparison of virus titers and infection rates, respectively, in Supplementary Fig. 2. All statistical tests were performed using GraphPad Prism 9.

### Reporting summary

Further information on research design is available in the Nature Portfolio Reporting Summary linked to this article.

### Data availability

RNA-seq data have been deposited in NCBI Sequence Read Archive under accession number PRJNA885496. ATAC-seq, ChIP-seq data, and Hi-C data were available from previous studies, deposited under Gene Expression Omnibus accession number GSE210270 (ATAC-seq and ChIP-seq) and GSE95797 (Hi-C). Source data for other graphs in the main and Supplementary Figs. are provided in Supplementary Data 2.

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

## Acknowledgements
We thank Benoit Besson, Finny Varghese, and other members of the laboratory for discussions. We thank Martijn van Hemert (Leiden University Medical Center), Sandra Junglen (Charité Universitätsmedizin Berlin), Beate Kümmerer (University of Bonn), Andres Merits (University of Tartu), Jason Rasgon (Pennsylvania State University), and Jolanda Smit (University of Groningen) for providing materials. The following reagent was provided by the NIH/NIAID Filariasis Research Reagent Resource Center for distribution by BEI Resources, NIAID, NIH: Ae. aegypti, strain black eye Liverpool, eggs, NR-48921. This work was supported by a VENI grant from the Dutch Research Council to P.M. (grant number VI.Veni.202.035) and a VICI grant from the Dutch Research Council to R.P.v.R. (grant number 016.VICI.170.090). The graphical abstract was inspired by the Super Mario Bros video game (Nintendo).

## Author contributions
Conceptualization, J.Q., R.P.v.R.; Investigation, J.Q., M.S., L.C., S.R.M., G.J.O., Formal analysis, J.Q., M.S.; Visualization, J.Q., M.S.; Writing – Original Draft, J.Q., R.P.v.R.; Writing – Review & Editing, J.Q., M.S., L.C., S.R.M., G.J.O., P.M. R.P.v.R.; Funding Acquisition, R.P.v.R.; Supervision, P.M., R.P.v.R.

## Competing interests
The authors declare no competing interests.
