## [Transparent Peer Review file · Communications Biology]

The Hsf1-sHsp cascade has pan-antiviral activity in mosquito cells

Corresponding Author: Professor Ronald Van Rij

Version 0:

Reviewer comments:

Reviewer #1

(Remarks to the Author)

The authors combined RNASeq, in vitro, in vivo and usage of chemicals to identify a new axis involving HSPs in mosquitoes that has broad antiviral function. The paper is overall clear and well supported. Nonetheless, I recommend the addition of two experiments to strength the scientific discovery before acceptance.

1. Quantification of CHIKV in Aag2 cells at the time points used to perform RNASeq. This information will help relate gene expression intensities to actual viral genomes which act as trigger.
2. quantification of sHsps in orally infected mosquito midguts supplemented or not with DTHIB. I understand the limitation of the previous chemicals (Hfsa and KRIBB11) in using them for oral feeding. However, the effect of DTHIB on sHsps expressions has to be shown to relate the proviral function of DTHIB to the Hsf1-sHsps axis. It is very possible that DTHIB has side effects unrelated to sHsps or Hsf1, even more so as it does not influence Hsf1 as expected. I suggested technical improvements to enable the authors to quantify gene expression in midguts early after blood feeding.

Please see below for minor comments:

Introduction

Clear. When introducing the immunity functions, the authors missed the finding that JNK in Aedes mosquitoes is a broad-spectrum antiviral pathway (efficient against DENV, ZIKV and CHIKV) acting through complement and apoptosis (Chowdhury et al, 2020. PLoS Pathogens). Please add this information.

Results

Sub-section 1:

The authors should quantify virus replication either by qPCR or titration to report the infection kinetics and how this relates to the DEgenes.

- I. 109. Although it's written in the figure legend, it would be clearer to mention the time post infection in the text.

Sub-section 2:

I. 115: It should be mentioned in the text how long after infection they quantified CHIKV.

L. 147: Either I don't understand the sentence or I do not agree. Inhibiting the Hsf1-sHsp cascade by addition of hfsa1 before inoculation and up to 4hpi inhibited virus replication. Hence, all steps of the viral cycle before 4hpi are potential targets and this include attachment/entry. Also, if the effect was targeting replication, we would expect the effect to continue after 4hpi as replication is continuously taking place. Please clarify or modify.

Sub-section 3:

L. 159: it is not clear whether the authors generated the RNAseq data or if they used data already available. Please clarify.

Sub-section 4:

I. 185: Of note, it is completely doable and has been done to quantify gene expression early after oral blood feeding. To do this, one can remove the blood bolus during dissection. The quantification of sHsp after administration of the new compound would be important to relate the antiviral function to the Hsf1-sHsp axis.

I. 190: Reduction of infection rate is suggestive of a function of Hsf1 in initiating infection by acting at early steps such as entry/attachment/release of the viral genome.

Discussion

I. 228: as mentioned above, to me, there is no evidence that the target of the Hsf1 axis is post-entry, even perhaps the opposite. Please clarify.

Methods

I. 253: mention where the BHK cells came from.

Figures:

Fig. 4A: the title for the y-axis is missing.

Reviewer #2

(Remarks to the Author)

This is a novel RNAseq study looking at expression of *Ae. aegypti* cells in response to CHIKV during the very early stages of infection < 48 hrs. Much of the changes in expression occurred in the later hours and involved HSP70 and 8 small HSPs. The role of sHSPs in viral infection is poorly understood. The set of 8 genes appear to be co-located in a coordinated expressed region. An examination of the promoters for these genes revealed binding domains in the transcription factor Hsf1. They were also marked by histone marks and poly 2 occupancy suggestive of active transcription. Silencing Hsf1 led to increased transcription of all 8 sHSPs. Knocking these genes down individually and as a group increased CHIKV loads. This was not the case for HSP70. A small molecule inhibitor was used that prevents recruitment of Hsf1 to binding sites, this led to increased viral loads as predicted. Several of the genes surprisingly showed increased expression, possibly in response to CHIKV. These effects also occurred for SINV and ASALV as well as for CHIKV in diverse mosquito cell species. The inhibitor & activator were added at different time points post infection, demonstrating the importance of the sHSPs early during infection.

This is a very systematic set of experiments aimed to dissect and demonstrate the involvement of the sHSPs in antiviral activity for arboviruses in cells after finding their enrichment in mosquitoes early during CHIKV infection. The effect sizes are also very strong, making for a convincing case. HSP70 serves as a comparator to demonstrate the distinct involvement of the sHSP. The local characterization of the chromosome and identification of the transcription factor, allowed for a range of powerful expression knockdown experiments with viral load as the readout. The paper is very well written. I am excited about this work and the field will be eager to see additional future work in mosquitoes relating to this pathway.

Comments:

There is a lot of data here! The figures require some time and study to understand what they are actually showing. It might be worth splitting them up into more figures and making them bigger for the reader to improve visibility. Currently, deciphering all the symbols and colors and small font is a challenge.

In the discussion can you hypothesize on what you think the gene specific tissue based differences in expression might mean for use of these sHSPs for virus control. Are some of these expression patterns more likely to be relevant to the process of infection? While flies are indeed different, what does fly Atlas show for frame of reference? What about other tissue specific expression studies in *Ae. aegypti*? Do these genes turn up in those screens? – if so, what are the factors that trigger their expression. Both the fly and mossier data (if it exists) may assist with understanding the function of these genes.

You go through the exercise of showing the NJ tree – that mainly points out that there are 2 distinct clusters of the sHSPs. This left me wondering about the radiation of these sHSP genes. Do all the mosquito species have 8 (and how many do *Drosophila* have)? Please add this to the discussion.

Reviewer #3

(Remarks to the Author)

In "The Hsf1-sHsp cascade has pan-antiviral activity in mosquitoes," Qu and colleagues identify heat shock factor 1 (Hsf1) as a regulator of eight small heat shock proteins (sHsp), and they find that this cascade acts as an early response against CHIKV infection. Through cell culture studies, the authors test how activation or inactivation of Hsf1 affects other viruses as well, and they conclude that the Hsf1-sHsp cascade is pan-antiviral. The manuscript is well written and presents rich experimental data to justify observations from an RNAseq time course study performed in cell culture. However, there are a few gaps I feel must be addressed before the manuscript is suitable for publication.

The most pressing concern is in regards to the in vivo work performed in the mosquito. The authors have shown in their cell culture work that the Hsf1-sHsp cascade acts as an early response to CHIKV (Figure 3) and that activating Hsf1 has a dramatic impact on CHIKV replication up until 8 hpi. However, data in vivo is only shown 3 days post infection in the mosquito, where the impact of activating Hsf1 was modest at best. Given the effect was early in the infection process in cells, and because CHIKV is known to replicate very quickly in *Ae. aegypti*, especially in comparison to flaviviruses such as DENV or ZIKV, I question why the authors did not look at CHIKV RNA levels earlier in the infection process in vivo? I know that the authors mention in lines 184-186 that it is technically challenging to collect midguts after a bloodmeal, but it is a pretty common practice in the field to clean the midguts of blood by dissecting them open and dipping them in PBS a few

times to release the blood bolus. Minor point that I also find the title of Figure 4 a little misleading because Figure 4B & Figure 4C were performed in vitro, even though the title says "antiviral activity of the Hsf1-sHsp axis in vivo." Final point on this matter: the authors looked at gene expression of the sHsps in different mosquito tissues 3 & 5 dpi CHIKV and found great variability/no impact on gene expression (Fig. S2). In the Discussion, the authors explained that this could be due to the fact that "the infection state of different tissues or body parts are highly variable," yet in Figures S2B & S2C, they have shown that all organs were infected. I think a better way to say this would be the levels of virus infection across the different tissues is likely different as opposed to the state of infection, since they are all infected. I wonder also if they could correlate the gene expression with levels of CHIKV RNA to advert this problem?

Is it possible to show a CHIKV growth kinetics curve in the Aag2 cells at the different time points to show how "early" v. "late" infection stages were established? I'm not clear on what it means. Is it pre v. post-eclipse stage? I think further explanation on this matter is necessary.

Did the authors silence hsf1 in mock infected cells (Figure 1E) to see if silencing the gene itself has any impacts on cell growth rate? Along these lines, maybe it would be good to check this by normalizing viral RNA levels by a housekeeping gene.

If HSP70 is known to be involved in the viral cell cycle (lines 96-97), why was there no affect on virus replication when it was silenced (Fig. 2C)? Was it the time point tested? Further discussion could help here.

The authors say that they observed re-repression or even activation of sHsps at 12 hpi, probably due to increased CHIKV replication (lines 125-128). Could the authors mock infect cells and treat them with the same inhibitor to avoid this issue and show that repressing HSF truly prevents gene expression of all sHSPs?

In Fig 2G, KRIBB11 treatment in DENV infection is missing even though cited in the text (135-136).

In Fig 2J, why is such a big difference still ns? What does each circle represent?

In line 195, I think the authors should add in the cells of 3 vector mosquito species.

Version 1:

Reviewer comments:

Reviewer #1

(Remarks to the Author)

Dear Editor,

I commend the authors for addressing my main concerns, especially about the effect of the DTHIB inhibitor in vivo. They have now generated new in vivo data which does not completely reproduce the in vitro observations. Although I understand that it is deceptive and may diminish the strength of the message, the data should be shared with the community for transparency and to prevent others from repeating it. I therefore recommend that the in vivo data be published as supplementary data and presented in the discussion. The in vivo data should include the viral RNA quantification as well as the gene expression analyses.

Best regards

Reviewer #2

(Remarks to the Author)

The authors have addressed all major concerns.

We thank the editor and reviewers for their constructive comments, useful suggestions, and chance to revise our manuscript, now entitled “*The Hsf1-sHsp cascade has pan-antiviral activity in mosquito cells*”. Please find below a point-by-point response to the reviewers’ comments.

Reviewer #1

The authors combined RNASeq, *in vitro*, *in vivo* and usage of chemicals to identify a new axis involving HSPs in mosquitoes that has broad antiviral function. The paper is overall clear and well supported. Nonetheless, I recommend the addition of two experiments to strength the scientific discovery before acceptance.

We thank the reviewer for the supportive comments.

Major comments

1. Quantification of CHIKV in Aag2 cells at the time points used to perform RNASeq. This information will help relate gene expression intensities to actual viral genomes which act as trigger.

CHIKV has been quantified in the samples used for RNA-seq via RT-qPCR and by mapping the RNA-seq data to the viral genome. These data are now shown in supplemental figure 1. As expected, we see a strong increase of CHIKV RNA copies from 4 hours after infection onwards (see also our response to Comment 2 of Reviewer 3)

2. Quantification of sHsps in orally infected mosquito midguts supplemented or not with DTHIB. I understand the limitation of the previous chemicals (Hfsa and KRIBB11) in using them for oral feeding. However, the effect of DTHIB on sHsps expressions has to be shown to relate the proviral function of DTHIB to the Hsf1-sHsps axis. It is very possible that DTHIB has side effects unrelated to sHsps or Hsf1, even more so as it does not influence Hsf1 as expected. I suggested technical improvements to enable the authors to quantify gene expression in midguts early after blood feeding.

We thank the reviewer for this important comment, which was also raised by Reviewer 3.

We performed the experiments proposed by the reviewers. We fed Ae. aegypti mosquitoes on CHIKV infected blood with or without DTHIB (100 mM). Mosquitoes were then homogenized or dissected at 1, 2, and 3 days post CHIKV blood feeding and relative gene expression was measured using RT-qPCR (as described in the Materials & Methods section of our previous submission). RNA expression of HSP70 and the sHsp genes did not significantly differ between all four experimental groups at all timepoints (an example of one sHsp gene is shown in Rebuttal Fig, 1A-C, below), suggesting that neither DTHIB treatment nor CHIKV infection induces the expression of sHsps genes at the chosen timepoints.

Interestingly, viral infection rates in the midgut (the proportion of virus infected mosquitoes) were significantly reduced at 3 days post infection in the presence of DTHIB (fig 1D), which is similar to our observations in Figure 4D of the original submission. However, infection rates at earlier time points and viral titers did not significantly differ between DTHIB treated and control mosquitoes.

To test whether DTHIB affects small heat shock gene expression at an earlier time point, we fed mosquitoes with blood with or without DTHIB (100 mM) and assessed gene expression at 12 hours post blood feeding. However, sHsp RNA expression was not significantly different between the samples with and without compound treatment at both timepoints (an example is shown in fig 1E).

*Together, these somewhat disappointing results indicate that DTHIB treatment does not seem to modulate sHsp or Hsp70 expression in adult mosquitoes. As KRIBB11 and hfsa1 were poorly soluble in blood, we therefore lack the tools to robustly modulate sHsp gene expression to study the antiviral activity of the Hsf1-sHsp axis *in vivo*. We therefore decided to delete all *in vivo* data (Figure 4 of the original submission) from the manuscript and solely discuss the *in vitro* data. Acknowledging this*

limitation of our study, we have been careful not to overstate our conclusions and explicitly indicate that the Hsf1-sHsp axis is antiviral in cells (e.g., in the title, abstract line 17, discussion lines 174 and 215)

While revising our manuscript, we reorganized the presentation of our data as follows:

- Figure 2D of the original manuscript has been moved to Figure 3 of the revised manuscript, along with newly generated data in response to Reviewer 3, Comment 5, Figure 2E-J of the original manuscript is now presented as Figure 5 of the revised manuscript.
- Figure 3, together with Supplemental Figure S1 of the original manuscript is now Figure 4.
- Figure 4 and Figure S2 (in vivo data) from the original submission have been deleted from the manuscript.

Rebuttal Figure 1.

A–C) RNA expression of a representative sHsp gene (*AAELO13350*) in mosquito midguts at 1 (A), 2 (B) and 3 (C) days after feeding with blood (dpb) in the presence or absence of DTHIB (100 mM) and/or CHIKV, as indicated. RNA expression is measured by RT-qPCR and presented as Δ Ct values of *AAELO13350* over the *LAP* housekeeping gene. Symbols reflect data of individual mosquitoes. **D)** CHIKV titers and infection rates in mosquito midguts at 1-, 2- and 3-days post blood feeding in the presence (purple) or absence (green) of DTHIB. Viral titers were determined using end-point dilution assays. Triangles represent titers in individual midguts and the bars indicate the mean. Symbols below the x-axis indicate samples in which viral titers were undetectable. Ratios below indicate the infection rates, presented as the number of positive over the total number of samples. Statistical significance was tested using a Chi-squared tests (* $p < 0.05$) **E)** *AAELO13350* RNA expression at 12 hours post blood feeding. Δ Ct values of *AAELO13350* over the *LAP* housekeeping gene of mosquito midguts 12 hours post blood feeding in the absence (red) or presence (blue) of DTHIB (100 mM). Please note that we only show the expression levels of a representative sHsp gene (of $n = 8$ tested). The trends for the other sHsp genes are similar to this gene, which we selected for visualization.

Minor comments:

3. When introducing the immunity functions, the authors missed the finding that JNK in *Aedes* mosquitoes is a broad-spectrum antiviral pathway (efficient against DENV, ZIKV and CHIKV) acting through complement and apoptosis (Chowdhury et al, 2020. PLoS Pathogens). Please add this information.

We apologize for this oversight. The reference is now cited on lines 57 and 180.

4. Sub-section 1:

The authors should quantify virus replication either by qPCR or titration to report the infection kinetics and how this relates to the DEgenes.

See our response to major comment 1. A time course of viral replication is now provided in supplemental figure 1.

Line. 109. Although it's written in the figure legend, it would be clearer to mention the time post infection in the text.

We agree and changes have been (line 112 and 114)

Line. 159: it is not clear whether the authors generated the RNAseq data or if they used data already available. Please clarify.

This is a partly a new generated RNA-seq dataset, see Materials & Methods line 285-287.

5. Sub-section 2:

Line 115: It should be mentioned in the text how long after infection they quantified CHIKV.

We agree and changes have been made (line 112 and 114)

Line 147: Either I don't understand the sentence or I do not agree. Inhibiting the Hsf1-sHsp cascade by addition of hsf1 before inoculation and up to 4hpi inhibited virus replication. Hence, all steps of the viral cycle before 4hpi are potential targets and this include attachment/entry. Also, if the effect was targeting replication, we would expect the effect to continue after 4hpi as replication is continuously taking place. Please clarify or modify.

Viral entry represents the initial step in a viral infection. If hsf1a would inhibit viral entry, which typically occurs within 1 hour post inoculation, we would expect the compound to inhibit viral replication when added before or during inoculation, but to lose its activity when added at a later time point. As our data indicate that hsf1a is active even when added 4 hours post-inoculation, we conclude that viral entry is not the target of hsf1a's actions.

Notably, our observations reveal a marked contrast in outcomes: a potent effect of drug administration when applied up to 4-hour post-infection, followed by a gradual loss of inhibition thereafter. Thus, it is plausible that the critical target events occur around the 4 hpi time point, suggesting an effect on early viral RNA replication. We have added a statement and reference on the time frame in which entry occurs (line 146-147) and hope that our line of argument is now clear.

6. Sub-section 4:

Line 185: Of note, it is completely doable and has been done to quantify gene expression early after oral blood feeding. To do this, one can remove the blood bolus during dissection. The quantification of sHsp after administration of the new compound would be important to relate the antiviral function to the Hsf1-sHsp axis.

Thank you for this suggestion. See our response to Comment 2 of this reviewer.

Line 190: Reduction of infection rate is suggestive of a function of Hsf1 in initiating infection by acting at early steps such as entry/attachment/release of the viral genome.

We have addressed this point in response to Comment 5.

7. Discussion:

Line. 228: as mentioned above, to me, there is no evidence that the target of the Hsf1 axis is post-entry, even perhaps the opposite. Please clarify.

See our response to Comment 5.

8. Methods:

Line. 253: mention where the BHK cells came from.

This information has now been provided (line 229).

9. Figures:

Fig. 4A: the title for the y-axis is missing.

As indicated in our response to Comment 2 of this reviewer, this figure is removed from the revised manuscript.

Reviewer #2

This is a very systematic set of experiments aimed to dissect and demonstrate the involvement of the sHSPs in antiviral activity for arboviruses in cells after finding their enrichment in mosquitoes early during CHIKV infection. The effect sizes are also very strong, making for a convincing case. HSP70 serves as a comparator to demonstrate the distinct involvement of the sHSP. The local characterization of the chromosome and identification of the transcription factor, allowed for a range of powerful expression knockdown experiments with viral load as the readout. The paper is very well written. I am excited about this work and the field will be eager to see additional future work in mosquitoes relating to this pathway.

We thank reviewer 2 for the enthusiastic comments.

Comments:

1. There is a lot of data here! The figures require some time and study to understand what they are actually showing. It might be worth splitting them up into more figures and making them bigger for the reader to improve visibility. Currently, deciphering all the symbols and colors and small font is a challenge.

We agree with the reviewer that the figures were information dense. In the revised version of the manuscript, we have restructured several panels (see Comment 2, Reviewer 1) and think that the figures are more intuitive now.

2. In the discussion can you hypothesize on what you think the gene specific tissue based differences in expression might mean for use of these sHSPs for virus control. Are some of these expression patterns more likely to be relevant to the process of infection? While flies are indeed different, what does fly Atlas show for frame of reference? What about other tissue specific expression studies in *Ae. aegypti*? Do these genes turn up in those screens? – if so, what are the factors that trigger their expression. Both the fly and mossier data (if it exists) may assist with understanding the function of these genes.

We agree that tissue specific expression is interesting and merits further investigation. However, as we have removed the in vivo data from the revised manuscript, we feel a discussion of tissue specific difference in sHsp gene expression is not useful anymore.

3. You go through the exercise of showing the NJ tree – that mainly points out that there are 2 distinct clusters of the sHSPs. This left me wondering about the radiation of these sHSP genes. Do all the mosquito species have 8 (and how many do *Drosophila* have)? Please add this to the discussion.

*This is an interesting point. The *Drosophila melanogaster* genome encodes 11 sHsp genes, some of which are clustered on a small region in the genome (like in *Ae. aegypti*) (<https://flybase.org/reports/FBgg0000507.htm>; <https://doi.org/10.3390/ijms19113441>). We expect that there may be variation in the number of Hsp genes in other mosquitoes, but given the high duplication level in the genomes of other culicine mosquitoes, estimates of the number of duplicated genes (like Hsp genes) will not be reliable.*

Please note that the phylogenetic tree is not rooted, nor did we statistically test for robustness of the clustering. We expect that the two clusters are merely suggested by the visualization and not statistically supported and therefore we chose not to discuss this in the manuscript. We therefore refrained from an in-depth discussion of the phylogenetic tree.

Reviewer #3

In "The Hsf1-sHsp cascade has pan-antiviral activity in mosquitoes," Qu and colleagues identify heat shock factor 1 (Hsf1) as a regulator of eight small heat shock proteins (sHsp), and they find that this cascade acts as an early response against CHIKV infection. Through cell culture studies, the authors test how activation or inactivation of Hsf1 affects other viruses as well, and they conclude that the Hsf1-sHsp cascade is pan-antiviral. The manuscript is well written and presents rich experimental data to justify observations from an RNAseq time course study performed in cell culture. However, there are a few gaps I feel must be addressed before the manuscript is suitable for publication.

We would like to thank reviewer 3 for the supportive comments.

Major comments

1. The most pressing concern is in regards to the in vivo work performed in the mosquito. The authors have shown in their cell culture work that the Hsf1-sHsp cascade acts as an early response to CHIKV (Figure 3) and that activating Hsf1 has a dramatic impact on CHIKV replication up until 8 hpi. However, data in vivo is only shown 3 days post infection in the mosquito, where the impact of activating Hsf1 was modest at best. Given the effect was early in the infection process in cells, and because CHIKV is known to replicate very quickly in *Ae. aegypti*, especially in comparison to flaviviruses such as DENV or ZIKV, I question why the authors did not look at CHIKV RNA levels earlier in the infection process in vivo? I know that the authors mention in lines 184-186 that it is technically challenging to collect midguts after a bloodmeal, but it is a pretty common practice in the field to clean the midguts of blood by dissecting them open and dipping them in PBS a few times to release the blood bolus. Minor point that I also find the title of Figure 4 a little misleading because Figure 4B & Figure 4C were performed in vitro, even though the title says "antiviral activity of the Hsf1-sHsp axis in vivo." Final point on this matter: the authors looked at gene expression of the sHsps in different mosquito tissues 3 & 5 dpi CHIKV and found great variability/no impact on gene expression (Fig. S2). In the Discussion, the authors explained that this could be due to the fact that "the infection state of different tissues or body parts are highly variable," yet in Figures S2B & S2C, they have shown that all organs were infected. I think a better way to say this would be the levels of virus infection across the different tissues is likely different as opposed to the state of infection, since they are all infected. I wonder also if they could correlate the gene expression with levels of CHIKV RNA to advert this problem?

We thank the reviewer for the valid comments. Reviewer 1 had similar concerns. We have performed the experiments suggested by the reviewers, but we were unable to verify that DTHIB affects sHsp gene expression in vivo. As a consequence, we are unable to functionally link the sHsps response to viral

infection in adult mosquitoes and chose to remove all in vivo data from the manuscript. Please see our response to Comment 2 of Reviewer 1 for further explanation.

2. Is it possible to show a CHIKV growth kinetics curve in the Aag2 cells at the different time points to show how "early" v. "late" infection stages were established? I'm not clear on what it means. Is it pre v. post-eclipse stage? I think further explanation on this matter is necessary.

We have established CHIKV growth kinetics in the samples used for RNA-seq by RT-qPCR (new figure Fig S1A) and quantification of viral reads in the RNA-seq data. As expected, viral RNA levels are rapidly increasing over time. These data are now shown in Supplemental Figure 1. Moreover, our time-of-addition experiments in which we quantified both negative and positive-sense viral RNA levels provide extra insights (Figure 4C): Here, we observed that the levels of negative-sense viral RNA starts to increase from 4 hpi onwards, suggesting that this reflects the onset of viral RNA replication.

Thus, the time points correspond to the initiation of RNA replication (4 hpi), active RNA replication phase (8 hpi) and advanced infection in which new particles are shed in the supernatant (48 hpi). In the manuscript, we retained the terms early and late response, although we acknowledge that these terms are a bit imprecise.

3. Did the authors silence hsf1 in mock infected cells (Figure 1E) to see if silencing the gene itself has any impacts on cell growth rate? Along these lines, maybe it would be good to check this by normalizing viral RNA levels by a housekeeping gene.

Visual inspection suggested that cell numbers were similar between control and Hsf1 knockdown cells. In addition to this, yields after total RNA isolation were very similar between control and Hsf1 knockdown cells, and Ct levels of mRNA from the LAP housekeeping gene (used for normalization of our RT-qPCR data) are stable upon hsf1 knockdown. Based on these data, we conclude that Hsf1 knockdown does not affect cell growth.

4. If HSP70 is known to be involved in the viral cell cycle (lines 96-97), why was there no affect on virus replication when it was silenced (Fig. 2C)? Was it the time point tested? Further discussion could help here.

To our knowledge, the role of Hsp70 has only been studied in the context of dengue and Zika virus infection in mosquito cells (this information has now been included in the manuscript, line 93). Moreover, we evaluated the effect of hsp70 only at 8 hpi, and we can therefore not formally exclude that Hsp70 may affect other stages of the viral life cycles, which we now acknowledge in the manuscript on lines 115-116. Irrespective, it remains striking that Hsp70, which is induced to similar levels as sHsp genes (Fig 1B), does not affect CHIKV replication, whereas sHsp genes do.

5. The authors say that they observed re-repression or even activation of sHsps at 12 hpi, probably due to increased CHIKV replication (lines 125-128). Could the authors mock infect cells and treat them with the same inhibitor to avoid this issue and show that repressing HSF truly prevents gene expression of all sHSPs?

This is a valid point. As suggested by the reviewer, we mock infected cells, treated them with all further used compounds, and analyzed sHsp gene expression. These analyses showed that KRIB11 inhibits expression of most sHsp genes at 12 hours post treatment, and that this effect was less pronounced at 24 hours post treatment. In contrast, Hsf1a and DTHIB consistently induced sHsp gene expression at 12 and 24 hours post treatment. As these data are not confounded by viral infection, we included these newly generated data as Figure 3A-B in the revised manuscript. The results are described on lines 120-126.

6. In Fig 2G, KRIB11 treatment in DENV infection is missing even though cited in the text (135-136).

Thank you for pointing this out. We have corrected the text (line 158-160).

7. In Fig 2J, why is such a big difference still ns? What does each circle represent?

This was indeed an error in the figure of our original submission. The difference is indeed significant (figure 5F of the revised manuscript). Each circle represents one replicate within an experiment; this is indicated in the legend (line 578).

8. In line 195, I think the authors should add in the cells of 3 vector mosquito species.

We agree and we adjusted our statement on line 170 of the revised manuscript. More generally, we explicitly indicate that the Hsf1-sHsp axis is antiviral in cells (e.g., in the title, abstract line 17, discussion lines 174 and 215)

We thank the editor and reviewers for assessing the revised version of our manuscript "*The Hsf1-sHsp cascade has pan-antiviral activity in mosquito cells*". Please find below our response to the remaining comments in blue, italic font.

Reviewer #1

I commend the authors for addressing my main concerns, especially about the effect of the DTHIB inhibitor in vivo. They have now generated new in vivo data which does not completely reproduce the in vitro observations. Although I understand that it is deceptive and may diminish the strength of the message, the data should be shared with the community for transparency and to prevent others from repeating it. I therefore recommend that the in vivo data be published as supplementary data and presented in the discussion. The in vivo data should include the viral RNA quantification as well as the gene expression analyses.

As requested by the reviewer and editor, we now include the data as Supplementary Figure S2 and the associated methods in the Materials and Methods section..

Reviewer #2

The authors have addressed all major concerns.

Thank you.